# Graphene-Based Flexible Strain Sensor Based on PDMS for Strain Detection of Steel Wire Core Conveyor Belt Joints

**DOI:** 10.3390/s23177473

**Published:** 2023-08-28

**Authors:** Pengfei Li, Zhijie Li, Hongyue Chen, Yunji Zhu, Dada Yang, Yang Hou

**Affiliations:** School of Mechanical Engineering, Liaoning Technical University, Fuxin 123000, China; lipengfei@lntu.edu.cn (P.L.); chyxiaobao@126.com (H.C.); zhu747937477@163.com (Y.Z.); stivals58@163.com (D.Y.); hyhbang9703@163.com (Y.H.)

**Keywords:** flexible strain sensor, graphene, polydimethylsiloxane siloxane, rope core conveyer belt

## Abstract

Because of their superior performance, flexible strain sensors are used in a wide range of applications, including medicine and health, human–computer interaction, and precision manufacturing. Flexible strain sensors outperform conventional silicon-based sensors in high-strain environments. However, most current studies report complex flexible sensor preparation processes, and research focuses on enhancing and improving one parameter or property of the sensors, ignoring the feasibility of flexible strain sensors for applications in various fields. Since the mechanical properties of flexible sensors can be well combined with rubber conveyor belts, in this work polydimethylsiloxane (PDMS) was used as a flexible substrate by a simple way of multiple drop coating. Graphene-based flexible strain sensor films that can be used for strain detection at the joints of steel cord core conveyor belts were successfully fabricated. The results of the tests show that the sensor has a high sensitivity and can achieve a fast response (response time: 43 ms). Furthermore, the sensor can still capture the conveyor belt strain after withstanding high pressure (1.2–1.4 MPa) and high temperature (150 °C) during the belt vulcanization process. This validates the feasibility of using flexible strain sensors in steel wire core conveyor belts and has some potential for detecting abnormal strains in steel wire core conveyor belt, broadening the application field of flexible sensors.

## 1. Introduction

With the advancement of technology, flexible strain sensors are preferred over silicon-based sensors due to their excellent flexibility and ductility [1,2,3,4,5]. Their diverse and flexible structural forms have led to a wide range of applications in medical and health detection [6,7], artificial intelligence [8,9], wearable devices [10,11,12], and robotic skin [13,14]. Flexible sensors in various types are used in a wide range of applications. For example, flexible pressure sensors detect changes in force or pressure on the surface of an object, flexible temperature sensors are used to monitor fluctuations in ambient temperature, and flexible chemical sensors are dedicated to detecting the chemical composition of gases or liquids. However, the most prevalent and critical are flexible strain sensors. Particularly noteworthy is the wide adaptability of flexible strain sensors, which can be applied to a variety of interfaces where pressure and strain need to be detected. Whether confronted with a hard surface, such as a metallic structure, or a soft interface, such as skin, flexible strain sensors are able to accurately capture small changes in strain. Even more remarkable is that it is conformal and is able to realize accurate fit and measurement [15,16,17,18]. Flexible strain sensors have drawn a lot of interest from a variety of industries as a result of this special advantage. Researchers are continuously expanding and developing the application scope of flexible strain sensors in order to further explore their application potential in a variety of fields, as well as investigating ways to further improve the sensitivity, stability, and reliability of these sensors [1,19,20]. Considering that flexible strain sensors frequently experience stretching and bending under operating conditions, sensitivity and reliability are crucial factors in determining whether flexible sensors are practical in a variety of fields [21,22,23]. By incorporating silver nanoparticles into elastic fibers, Lee et al. [24] created a highly stretchable, large-strain flexible sensor that was successfully fitted into a glove to drive a manual robot, showcasing its potential for use in fields including wearable electronics. That sensor is capable of reaching an extremely wide strain sensing range of 450% deformation and has a high durability rating of more than 10,000 stretching cycles. Flexible strain sensors have potential for use in wearable electronics and biomedical engineering because of the wide strain range that enables successful integration into gloves to control manual robots and successful application to the human bladder system for health monitoring. Lin et al. [25] proposed a multifunctional flexible strain sensor with a low detection limit of 1% strain and a high stretchability of up to 217%. In contrast to most flexible sensors, which are limited in practical applications due to their low stretchability and sensitivity, its sufficiently wide strain range allows for sensors to be used in both human health and sports. Wang et al. [26] created a wearable flexible strain sensor with carbonized silk fabric that has a robust and incredibly high strain sensing range of up to 500% of the maximum stretch range. The sensor has the ability to be assembled as a wearable device. Unlike other flexible sensors, it exhibits extremely high sensitivity over a wide range of strains so that it can be used to detect both significant and subtle human activities, showing great potential for detecting a wide range of strains at the same time applicable in a number of fields.

In the current report, flexible sensors are widely used in many different fields, and extensive research has been conducted, particularly in the application of wearable technology and human information monitoring. In contrast, in this work, we propose a new application area for flexible sensors, namely strain detecting for steel wire core conveyor belts to identify abnormal strains in the rubber matrix as well as in the internal steel cord. Belt conveyors are a kind of mechanical equipment widely used in large-scale continuous transportation; the steel wire core conveyor belt is its main load-bearing component. The conveyor belt consists of internal steel cord and rubber matrix. The steel wire core conveyor belt in a conveyor is not a complete monolith but is joined together by joints. These joints are the most vulnerable parts of the belt. When subjected to external loads, the bonds between the steel cords and the rubber in the joints of a belt are more prone to accumulate damage. When large deformations occur in the joint area, some of the cords may loosen and pull away from the rubber matrix. When the joint is loosened and then loaded again, the damage increases, and in severe cases, the belt tears [27,28]. A number of preventive and corrective measures were proposed in response to the problem of conveyor belt damage in order to prevent the occurrence of belt breakage and detect the wire rope twitching inside the belt in a timely manner. Many preventive and corrective measures, such as protection devices that use X-ray images to predict twitching of wire-core conveyor belt joints and various types of belt-break catching devices, were proposed in recent research [29,30,31]. Lv et al. [32] proposed a technique for detecting longitudinal tears in conveyor belts by line laser, which can effectively eliminate the effect of variations in ambient light while also retaining the characteristics of longitudinal tears. The technique offered better real-time performance and high accuracy. Wang et al. [33] proposed a detection method based on sound recognition of longitudinal tearing of conveyor belts. Through the preprocessing algorithm to process the sound signal, a convolutional neural network model of the characteristics of the conveyor sound signal was established to recognize and detect the sound from longitudinal tearing of conveyor belts, and the accuracy rate achieved was as high as 94%, while the processing time was shortened to only 26.6 ms. That method has a wide detection range as well as high accuracy in recognizing whether a conveyor belt has produced a longitudinal tear. In terms of identifying damage and longitudinal tearing at the joints of conveyor belts, Yang et al. [34] used infrared spectroscopy to warn of longitudinal tearing of conveyor belts, and the experiment successfully collected precursor information about the tearing process of conveyor belts. Although there were many measures and methods to detect the joints of conveyor belts, there are still time consuming and less operable problems in recognizing the damage of conveyor belts subjected to abnormal strains. 

This study investigated the viability of flexible strain sensors used in steel wire core conveyor belts, to broaden their application in engineering fields, and to monitor the working efficiency and dependability of the sensors after being exposed to high pressure and high temperature. Using a quick preparation procedure, we were able to successfully fabricate a flexible graphene-based strain sensor. After performance and sensitivity tests, it was applied to conveyor belt joints via embedding the sensor film into the belt. During manufacturing steel wire core conveyor belts, the process of vulcanizing experienced high pressure and high temperature. The graphene-based flexible strain sensor is flexible, malleable and conformal. In addition, the softness of the rubber layer of the conveyor belt provides a good cushioning effect for the sensor inside. Therefore, it is possible to apply flexible strain sensors to the interior of steel wire core conveyor belts. Detecting abnormal strains in conveyor belts by flexible sensors will help to improve the reliability of steel wire core conveyor belt joints. At the same time, it is possible to understand the performance of flexible strain sensors after withstanding higher pressure and high temperature, and the results provide some support and reference for the application of flexible strain sensors in other engineering fields.

The fabrication process of the graphene-based flexible strain sensor and the obtained sensor morphology are described in detail in the Section 2. In the Section 3, we tested the sensing characteristics and reliability of this sensor and presented its embedding in wire rope core conveyor belts. Following that, experimental tests were carried out, and the results were analyzed. Finally, in the Section 4, this study and the work conducted are summarized.

## 2. Materials and Methods

### 2.1. Experimental Materials

In this experiment, monolayer graphene powder used as the modifying material for the flexible strain sensors was provided by Nanjing XFNANO Material Science and Technology Co., Ltd. (Nanjing, Jiangsu Province, China). The diameter of graphene powder was 0.5–5 μm, the maximum thickness was 0.8 nm, and the monolayer rate was 80%. Polydimethylsiloxane (PDMS) was used as the matrix material for the flexible strain sensors. Anhydrous ethanol was used as the dispersant of graphene powder. The mass ratio of PDMS (Sylgard 184 silicone elastomer base; Dow Corning) to curing agent (Sylgard 184 silicone elastomer curing agent; Dow Corning) was 10:1. They were mixed and stirred for 5–10 min before usage. The main materials required in the production of steel wire core conveyor belts are surface rubber (RIT Unvulcanized Top Rubber; UPM), core rubber (RIT Unvulcanized Core Rubber; UPM), and vulcanizing agent (SK823 Hot Vulcanizing Agent; UPM).

### 2.2. Preparation Process of Graphene-Based Flexible Strain Sensor Films

Figure 1 shows the whole production process for graphene-based flexible strain sensor using PDMS as matrix material. First, a certain amount of monolayer graphene powder was weighed using an electronic balance and mixed with anhydrous ethanol. The mixture was stirred uniformly and then ultrasonicated for 30 min to ensure that the graphene powder could be evenly dispersed in the anhydrous ethanol (Figure 1a). Next, the graphene dispersion was uniformly applied to the PDMS by drop coating after first covering a layer of PDMS and curing agent mixture on the pre-prepared mold (Figure 1b). That PTFE mold allows for easier release of the flexible sensor after subsequent curing. The molds with graphene/PDMS were then dried for a period of time, left to evaporate from the anhydrous ethanol, and the molds were coated with a layer of PDMS again (Figure 1c) to complete one cycle of the drop coating operation. Repeat the drop coating of graphene dispersion with PDMS pre-solid for 3–5 cycles. Next, the molds with graphene or PDMS were placed in a vacuum oven, and after removing air bubbles, they were heated at 75 °C for 2 h to fully cure the PDMS (Figure 1d) [35]. Finally, the graphene-based flexible strain sensor film was obtained by cutting the sensor size and preparing the electrodes after demolding (Figure 1e). In this study, three flexible strain sensor films with various graphene contents were created. By varying the amount of graphene in the flexible sensors, the performance of the flexible strain sensors can be quickly and easily adjusted. We controlled the ratio of graphene components to PDMS pre-solid by dropping graphene dispersion with a concentration of 0.5 mg/mL into the mold each time in order to compare the effects of various graphene contents on the sensitivity of flexible strain sensors in conveyor belts. Finally, three sets of flexible strain sensors with graphene component contents of 0.5 mg/mL, 1.0 mg/mL, and 1.5 mg/mL, respectively, were obtained.

### 2.3. Thin Film Morphology of Graphene-Based Flexible Strain Sensors

Figure 2a shows a physical picture of a graphene-based flexible strain sensor film. The sensor consists of only two parts, the main body part and the copper foil electrodes on both sides of the flexible sensor. In order to subsequently match the structural dimensions of the joints of the wire rope core conveyor belt. We cut the graphene-based flexible material obtained from the fabrication into 15 mm × 10 mm square cubes after fitting the electrodes, and Figure 2b,c illustrate the area of the effective sensitive layer of the sensor as 13 mm × 10 mm. In addition, the thickness of the sensor film was measured to be only 1 mm, and we deliberately made the flexible strain sensors as thin as possible in order to minimize the bulging of the rubber layer of the conveyor belt after the vulcanization process when it is subsequently applied to the steel cord conveyor belt. As can be seen in Figure 2d, the sensor is very flexible and has a high bending capacity. In addition, in Figure 2e, we stretched the uncut laminated electrode graphene/PDMS flexible material with a maximum stretching range of up to 150%. It shows that the flexible material used to fabricate the sensor has good tensile properties.

## 3. Results and Discussion

Figure 3 shows the experimental bench used to test the performance of a flexible strain sensor. The fixtures on the bench can be changed to indenter-type fixtures or clamp-type fixtures to allow for squeezing forces to be applied to the transducer film during testing and are also suitable for subsequent stretching of wire rope applied to wire rope core conveyor belts. The LCR resistivity meter (TH 2810B; Changzhou Tonghui Electronics Co., Ltd., Changzhou, Jiangsu Province, China) was used to detect the resistance parameters of the flexible strain sensors. The application of tensile and compressive forces to the flexible strain sensors was realized by means of a digital push-pull tester HLD (supplied by Yueqing Aidelberg Instrument Co., Ltd., Yueqing, Zhejiang Province, China). The variation range of its measurement is 0–1000 N, and the measurement accuracy is 0.001 N. As shown in Figure 3a, during the performance test of the sensor, the flexible strain sensor was fixed on the experimental circular table with adhesive tape so that the copper foil electrodes at both ends could be naturally pulled out to facilitate the connection to the measuring instrument. During the sensor performance test, the applied pressure varied from 0 to 800 kPa, and a series of repeated cyclic loading tests were performed to verify the sensitivity and reliability of the flexible sensor when subjected to pressure. Figure 3b shows a localized, enlarged view of the flexible strain sensor subjected to pressure testing.

### 3.1. Pressure Sensing Performance of Graphene-Based Flexible Strain Sensors

Before the flexible strain sensor film was vulcanized together with the wire rope core conveyor belt, we experimentally evaluated the sensitivity and reliability of three sets of flexible strain sensor samples with graphene component contents of 0.5 mg/mL, 1 mg/mL, and 1.5 mg/mL, respectively. The sensitivity performance S of the flexible sensor is evaluated as the amount of resistance change under a certain pressure, which is calculated as follows:S = (∆R/R_0_)/P(1)
where ∆R is the change in resistance when the sensor produces strain (∆R = R *−* R_0_), R_0_ is the initial resistance before strain, and P is the pressure applied to the sensor. We applied pressure to the flexible strain sensor film using a circular indenter in the test bench, and Figure 4 demonstrates the overall pressure sensing characteristics of the graphene-based flexible strain sensor film. In Figure 4a, the correlation between the change in resistance of the flexible strain sensor films with three different graphene composition contents and pressure is shown for a gradual increase in pressure from 1 to 600 kPa. The findings unmistakably demonstrate that sensor sensitivity rises as the proportion of graphene increases. The pressure range over which flexible strain sensors with various graphene compositions produce a large response can be seen to vary. Flexible strain sensors with a graphene component content of 0.5 mg/mL demonstrated a significant variation in resistance response in the range of 40 kPa, while flexible sensors with a graphene content of 1.0 mg/mL demonstrated a significant variation in resistance in the range of 57 kPa. Flexible strain sensors with a graphene component content of 1.5 mg/mL demonstrated a greater response range of 101 kPa. The sensitivity reached close to −0.0062 kPa^−1^ over the large response range. In addition, the flexible sensors exhibited a wide measurement range. Given that among the three sets of flexible sensors with different graphene fractions, the one with a graphene fraction of 1.5 mg/mL had the largest pressure range and the highest sensitivity to produce a large response, we chose to use this graphene fraction of the flexible strain sensor film for the subsequent testing and application in wire rope core conveyor belts. The sensor’s characteristics are shown in Figure 4b when it is dynamically loaded and unloaded at various pressures. We applied pressure to the sensor in increments of 50 kPa, 100 kPa, 150 kPa, and 200 kPa. It is clear from the figure that the sensor exhibits some regularity and stability. And as the pressure rises, the resistance changes more. But it is important to note that during dynamic loading and unloading under the same pressure action, the sensors’ hysteresis characteristics lead to an error in their pressure detection. The sensor’s dependability may be more significantly impacted by this. As shown in Figure 4c, continuous pressure cycles were applied to the transducer in the pressure range of 0 to 250 kPa to check the piezoresistive response curve of the transducer under the same loading and unloading conditions. This was done in order to observe the hysteresis characteristics of the transducer and confirm its reliability during dynamic loading. The experimental results indicate that there is a certain hysteresis characteristic of the sensor resistance change, but the figure demonstrates that the error that results is not significant and that the resistance change is not readily apparent during the loading and unloading process. This demonstrates the flexibility and dependability of the manufactured graphene-based flexible strain sensor film during dynamic loading and unloading. Figure 4d illustrates a small portion of the response curve of the sensor when the pressure is 100 kPa, as well as the relaxation time. The response speed and relaxation time of the sensor are critical for accurate detection, and if the response is too slow or the relaxation time is too long, it will result in biased sensor detection results. From the figure, it can be seen that both the response and relaxation times of the sensor are 43 ms, which is comparable to the response time of the sensor reported in some previous studies [36,37]. This indicates that the flexible strain sensor has a good and relatively fast response capability in practical applications. In addition, for the working environment of the wire rope core conveyor belt, we hope that the fabricated flexible strain sensor has good durability and can work stably for a long time. In order to verify the reliability of the sensor after a long period of operation and multiple pressure applications, in Figure 4e, we evaluated the durability of the flexible sensor by loading it with a pressure of 100 kPa several times and loading it with 300 consecutive cycles. The results show that the sensor maintains reasonable stability during repeated pressurization, and there is no significant degradation of the sensitivity. Meanwhile, it can be observed that the resistance change curves of the sensors are more similar in two different periods: after the beginning of the pressurization and after a long period of cyclic pressurization. In Figure 4f, the correlation between pressure and resistance of the flexible strain sensor during the loading of 0–600 kPa pressure after 1, 300, and 500 bends is demonstrated. It can be observed that there is no significant change in the resistance response of the sensor even after 500 bends. The two sets of test results in Figure 4e,f illustrate that the flexible sensors maintain excellent durability after many cycles of pressure application or repeated bending operations, and also show that the flexible sensors have good stability after a long period of operation. This point provides some support for the application of flexible strain sensors in wire rope core conveyor belts.

In addition to stress tests, we examined the characteristics of flexible sensor films’ performance under bending. To see the resistance change and response during bending, we placed the sensors on the wrist and the joint of the index finger. The resistance change characteristics of the sensor at the wrist joint and index finger are shown in Figure 4g,h. The findings unmistakably demonstrate that the resistance change ∆R/R_0_ of the sensor also exhibits periodic changes when the wrist and index finger joints are bent at periodic angles of 30° and 90°. The degree of resistance change of the sensor also increases with increasing bending angles. This can be explained by the fact that during bending, the sensor is stretched and its length relatively increases. As a result, the resistance change ∆R/R_0_ has a greater magnitude. The sensor’s deformation and the resistance change ∆R/R_0_ both decrease as the wrist and finger joints return to their initial states. This result shows that the flexible sensor can exhibit good sensitivity and responsiveness when subjected to bending. And our proposed flexible sensor has some potential for human body information detection. In addition, considering that the wire-core conveyor belt needs to run through the roller in practice, it is required that the flexible strain sensors have a good bending response if they are applied inside the conveyor belt. This also means that the sensor must be able to maintain stable performance over many bends and torsions. Therefore, ensuring that the flexible sensor has a good bending response is critical for the application of the sensor in wire rope conveyor belts.

### 3.2. Application of Flexible Strain Sensor Film in Wire Cord Conveyor Belt

In the current report, flexible strain sensors have been widely used in various fields, and most of the research work focuses on the application of wearable flexible strain sensors and human information detection. In this work, we envision the application of the proposed flexible strain sensors in wire rope core conveyor belts. The excellent properties of our proposed and fabricated graphene-based flexible strain sensor films, such as high sensitivity and faster response, were verified in previous tests. In addition, we consider the excellent conformality of the sensor and its ability to withstand large strains. Therefore, it becomes possible to apply it to the inside of steel cord core rubber conveyor belts. In a steel cord conveyor belt, the wire rope joints of the belt are the most fragile part of the belt, and when the belt is subjected to strain and damage, the wire rope at the internal joints will be the first to twitch, so we place flexible strain sensors in the joints inside the steel cord conveyor belt to detect and perceive the amount of pressure exerted on the conveyor belt and the strain exerted on the wire rope joints inside the belt.

Figure 5 illustrates a schematic diagram of the structural position of the graphene-based flexible strain sensor film in a wire rope core conveyor belt. As shown in the figure, according to the structure and production process of the wire rope core conveyor belt, the top and bottom layers of the conveyor belt specimen are the upper and lower unvulcanized face rubber, respectively, and another layer of unvulcanized core rubber needs to be arranged below the face rubber. The wire rope joints are arranged between the core rubber layers and covered by the upper and lower core rubbers. The graphene-based flexible strain sensor film we fabricated is located between the wire rope and the lower core adhesive, which is directly affixed to the wire rope inside the conveyor belt to better detect the resistance change of the sensor when the wire rope is pumped. It should be pointed out here that in order to avoid direct contact between the copper foil electrode of the sensor and the steel wire rope to form a conductive path, when arranging the flexible strain sensor, we took the measure of encapsulating the copper foil electrode surface of the sensor by spraying PTFE in advance to avoid large interference. Figure 6 shows a physical view of a wire rope core conveyor belt arranged with a flexible strain sensor. Figure 6a shows the actual structure inside the conveyor belt, where it can be seen that the sensor is underneath and close to the wire rope and that the sensor electrodes and wires protrude to facilitate subsequent connection of the specimen to an external measuring instrument. Figure 6b shows an unvulcanized wire rope core conveyor belt covered with a rubber layer containing a sensor film. Here, we place the unvulcanized conveyor belt with sensors in an aluminum alloy mold. The purpose of using the mold is to prevent the conveyor belt rubber from deforming and stretching too much during the vulcanization process and to avoid tearing of the internal flexible sensors in response to the deformation of the rubber. On a plate vulcanizing machine, the conveyor belt vulcanization process was carried out. As shown in Figure 6c,d, the unvulcanized conveyor belt containing the sensor and the mold is placed inside the plate vulcanizing machine. The vulcanization temperature and pressure are kept at 140 degrees Celsius and 1.3 MPa, respectively, and the vulcanization operation is completed with a 25 min holding pressure duration. A physical drawing of the wire rope core conveyor belt with a flexible graphene-based strain sensor film obtained by demolding at the conclusion of the vulcanization process is shown in Figure 6e.

Using the experimental test rig, we applied pressure to the surface of a vulcanized steel cord conveyor belt and simply stretched the steel cord inside the belt. In both cases, we evaluated whether the flexible sensors inside the belt could respond to the pressure applied to the belt and the slightest twitching of the wire rope inside the rubber, in order to verify whether the flexible sensor film could still work properly after being subjected to a large amount of pressure during the vulcanization process of the conveyor belt, as well as the feasibility of applying the flexible sensors inside the wire-core conveyor belts. Figure 7 illustrates two experimental approaches performed on a wire rope core conveyor belt containing a graphene-based flexible strain sensor film on an experimental bench and the resulting resistance change response curves. As shown in Figure 7a, it can be seen that the specimen of the wire rope core conveyor belt containing the flexible strain sensor film is placed on the test bench, and the circular indenter is used to apply pressure to the part of the conveyor belt where the sensor is placed, and the graph shows the curve of the pressure applied to the conveyor belt and the curve of the resistance response of the internal sensor, the time of change in the resistance of the sensor coincides with the time when the conveyor belt is subjected to the pressure, and the flexible sensor inside the conveyor belt shows a good response to the pressure applied to the conveyor belt. In Figure 7b, we replaced the clamping type fixture that held the conveyor belt’s rubber end and the other end that held the wire rope extending from the conveyor belt specimen. When a tensile force is applied to the wire rope in the conveyor belt, the internal sensor successfully captures the signal and causes the resistance to change. The resistance response curve shows that resistance changes during loading; however, the graph shows that the resistance change does not return exactly to the value of the original curve, but is slightly higher than the baseline resistance. The characteristics of the rubber material inside the steel cord conveyor belt can be used to explain this phenomenon. The rubber deforms when a force is applied, changing the resistance. When the force is released, the rubber slowly springs back into shape rather than instantly going back to its original shape. As a result, the rubber inside the conveyor belt continues to deform slightly after each unloading, producing resistance that is a little bit higher than the baseline resistance. This residual change slightly above the baseline resistance is due to the elastic deformation of the rubber material after a force has been applied. It takes time for the rubber material to fully recover, which causes the sensor resistance to change above the baseline value for a short period of time. This demonstrates that the flexible sensor inside the belt can detect actual changes in the belt and fully validates the feasibility of using the sensor on steel cord conveyor belts.

## 4. Conclusions

In summary, we successfully fabricated a graphene-based flexible strain sensor film using a simple preparation process of drop coating and applied it to a steel cord core rubber conveyor belt in order to detect the strain of the belt and the internal steel cord. Prior to practical application, the sensor was briefly tested and found to exhibit a series of excellent characteristics under pressure and bending, including high sensitivity (maximum sensitivity of −0.0062 kPa^−1^ under a wide pressure response range of 101 kPa), good linear response, and good reliability and durability. These excellent properties make the application of the sensors on conveyor belts possible. Subsequently, we designed the flexible strain sensors to be applied to conveyor belts and arranged the sensors at suitable locations on the belts. Meanwhile, the feasibility of graphene-based flexible strain sensor film inside the wire rope conveyor belt was demonstrated by making a specimen of the wire rope core conveyor belt and experimentally verifying it. The experimental results show that the sensor is not damaged by the huge pressure generated by the vulcanization process when making the conveyor belt because the buffer of the rubber layer in the conveyor belt plays a certain role. After the vulcanization process, the flexible sensor inside the conveyor belt can still work normally, and when pressure is applied to the surface of the conveyor belt or the wire rope in the conveyor belt is stretched, the flexible sensor is able to capture the corresponding force signals, and the resistance change curve of the sensor is basically consistent with the curve of the conveyor belt under force. The application of the sensor in the conveyor belt also proves that it can still work normally after withstanding large pressure, and at the same time, it provides a new method for detecting abnormal strain at the joints of the conveyor belt, which broadens the application field of the flexible sensor.

## Figures and Tables

**Figure 1 sensors-23-07473-f001:**
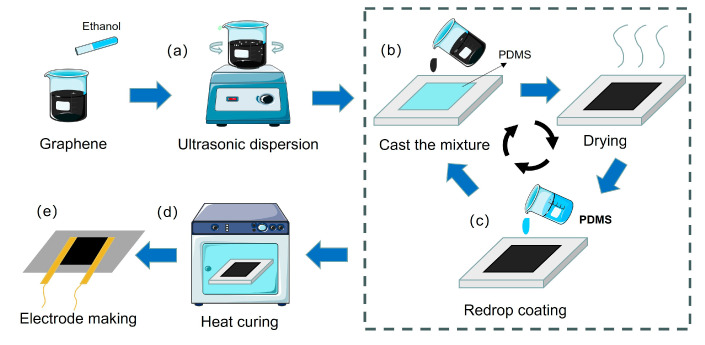
Fabrication of graphene-based porous flexible strain sensor films: (**a**) Ultrasonication for uniform dispersion of graphene (**b**) Graphene dispersion dropped onto PDMS-coated molds (**c**) After drying, re-coating with PDMS (**d**) Heat curing (**e**) Installation of electrodes after demolding.

**Figure 2 sensors-23-07473-f002:**
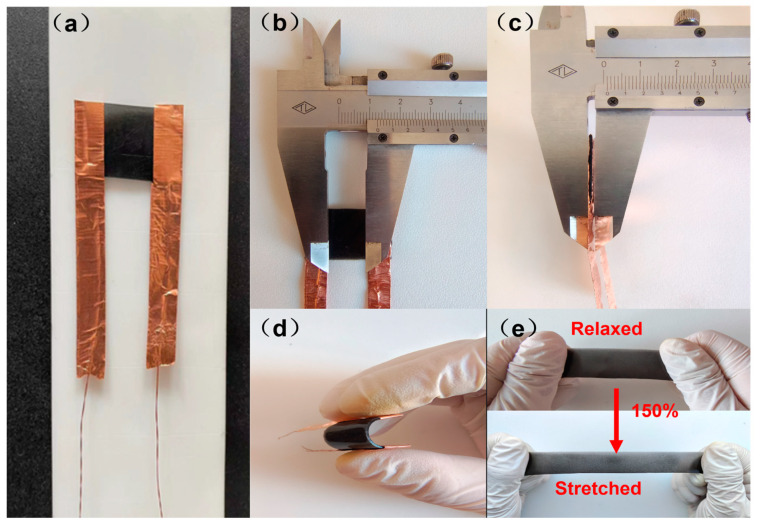
(**a**) Physical image of a graphene-based flexible strain sensor film (**b**,**c**) Effective sensitive layer size of the sensor and actual thickness of the film (**d**) Bendable properties of graphene-based flexible strain sensor films (**e**) Stretchable properties of graphene-based flexible sensing materials.

**Figure 3 sensors-23-07473-f003:**
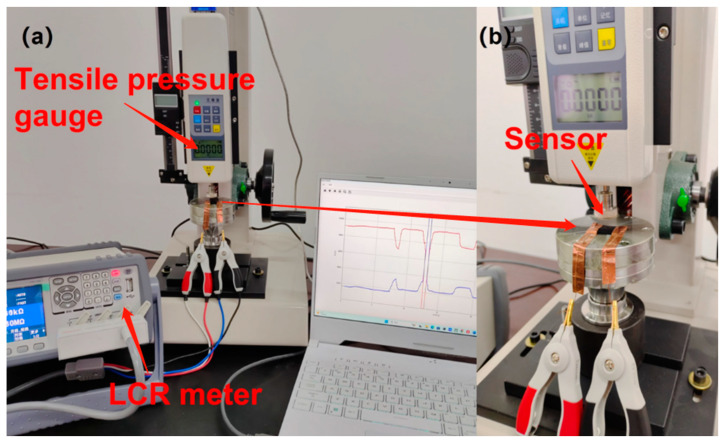
Experimental setup for sensor testing: (**a**) Sensor pressure test chart (**b**) Localized enlarged view of (**a**).

**Figure 4 sensors-23-07473-f004:**
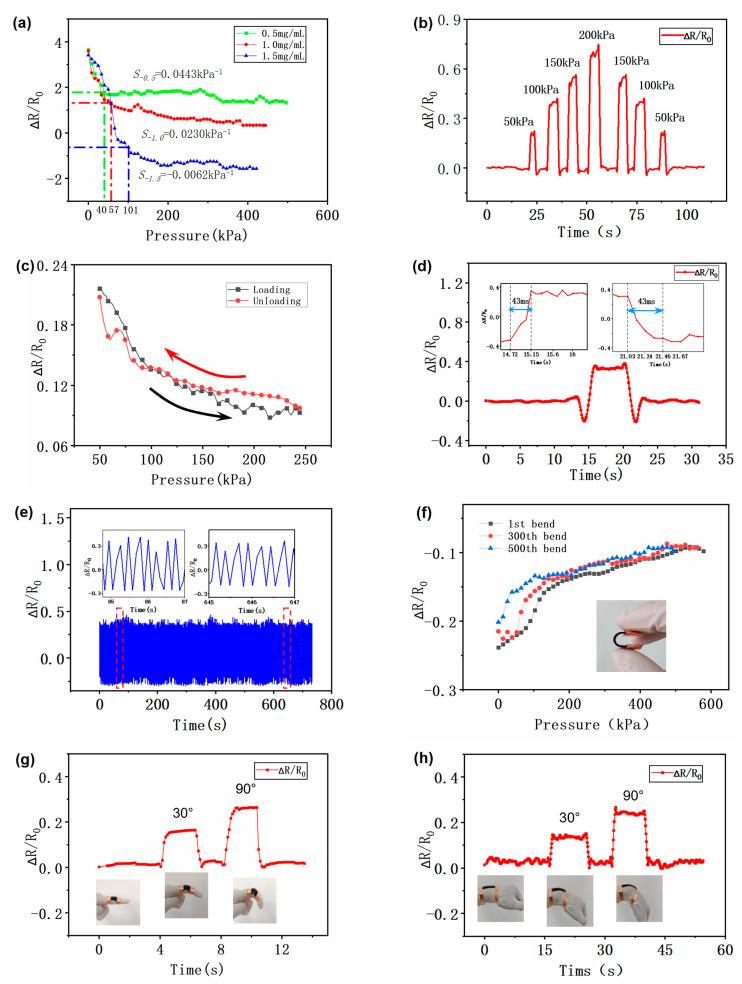
Performance testing of graphene-based flexible strain sensor films: (**a**) Pressure response curves of three sets of flexible strain sensor films with different graphene composition contents (**b**) Resistance response curves during dynamic loading/unloading of different pressures (**c**) Response curves of resistance in consecutive identical loading/unloading cycles (**d**) Sensor response speed and relaxation time (**e**) Resistance change curve under the same pressure for 300 consecutive repeated loadings (**f**) Resistance change curve of the sensor after bending 1 time, 300 times, and 500 times and applying the same continuous pressure (**g**,**h**) Changes in the resistance response of the sensor when flexed at 30° and 90° at the finger joints and wrist, respectively.

**Figure 5 sensors-23-07473-f005:**
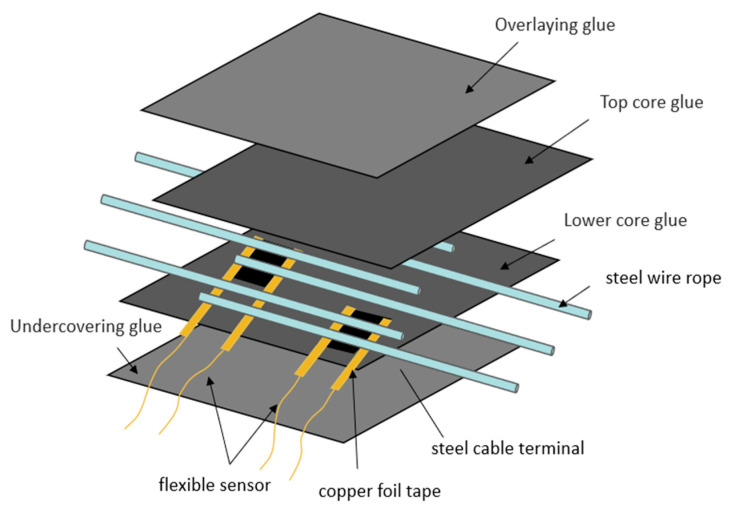
Structure of a steel cord conveyor belt with a graphene-based flexible strain sensor film.

**Figure 6 sensors-23-07473-f006:**
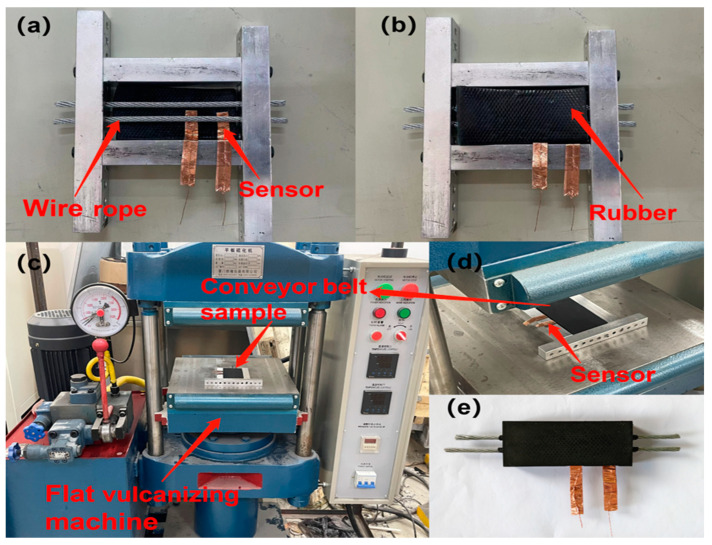
Vulcanization process of steel cord conveyor belts with sensors and physical diagrams: (**a**) The actual internal structure of the specimen of steel cord conveyor belt (**b**) Unvulcanized steel cord conveyor belt specimen with built-in flexibility sensor (**c**) The actual vulcanization process of steel cord conveyor belt (**d**) Localized enlarged view of figure (**c**); (**e**) Physical drawing of wire rope conveyor belt with flexible sensor.

**Figure 7 sensors-23-07473-f007:**
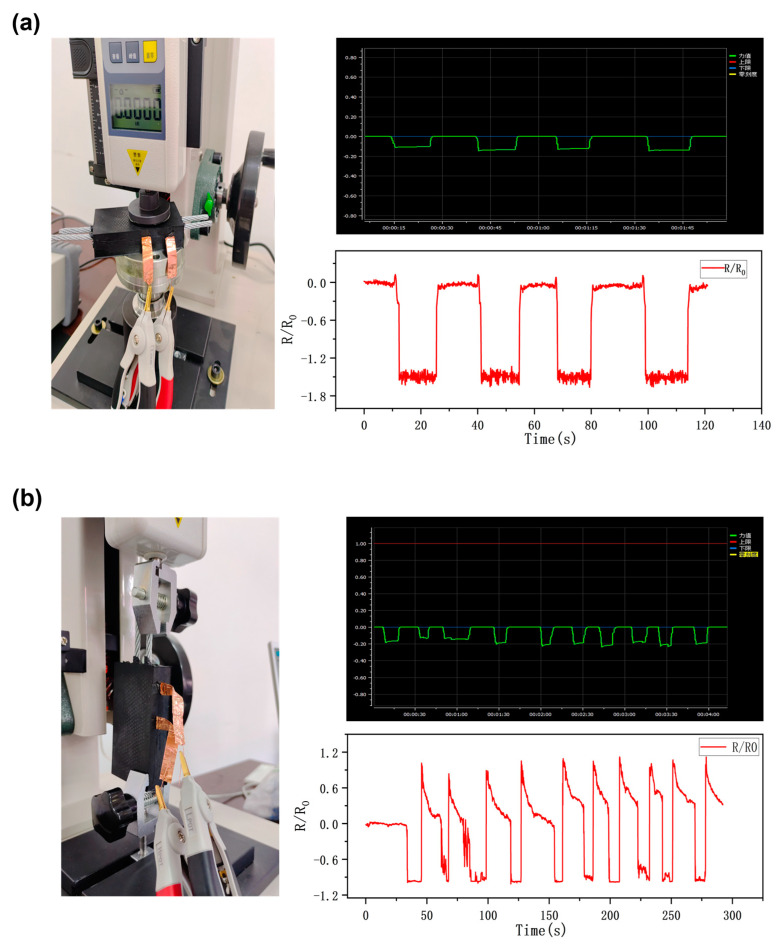
Sensor sensitivity for applications inside steel cord conveyor belts: (**a**) Response curve of wire rope core conveyor belt to pressure and resistance change of internal flexible sensor. (**b**) Response curve of the change in resistance of the internal flexible sensor and the amount of pulling force on the conveyor belt rope.

## Data Availability

No new data were created or analyzed in this study. Data sharing is not applicable to this article.

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
