# Peer review of "Graphene-Based Flexible Strain Sensor Based on PDMS for Strain Detection of Steel Wire Core Conveyor Belt Joints"

_sensors, 2023, doi:10.3390/s23177473_

Round 1
Reviewer 1 Report
1. In abstract (In this study, we used polydimethylsiloxane (PDMS), a common flexible substrate material, to create graphene-based flexible strain sensor films that can be used at the joints of steel wire rope core conveyor belts using a simple drop coating method. ) is not clear. Is need to rewrite as proposed method. Then explain the martial and testing system.
2. In introduction needs to add more details of reference [15-18].
3. In introduction few sentences (By embedding Ag nanoparticles in stretchable fibers, Lee et al. [24] created a highly stretchable large-strain flexible sensor for use in flexible sensor applications. ), (And the steel cord rubber conveyor belt is his main component; the conveyor belt consists of internal steel cord and rubber. ), (The method of detection has a larger detection range and higher accuracy. ) and (And the steel cord rubber conveyor belt is his main component; the conveyor belt consists of internal steel cord and rubber. )
4. Figure 2c and d is not clear , is better to find another way to show the sample size.
5. In section 2.1. Preparation process of graphene need to support your work with references and show why to select this parameters such as ( heated at 75°C for 2 h, dropping graphene dispersion with a concentration of 0.5 mg/m, three sets of flexible strain sensors with graphene component contents of 0.5 mg/mL, 1.0 mg/mL, and 1.5 mg/mL, )
Extensive editing of English language required
Reviewer 2 Report
In this work, authors have reported “Graphene flexible strain sensor based on PDMS for steel wire core conveyor belt connector strain detection”, where they proposed a strain sensor and its practical applications. The manuscript seems well described and presented but needs to address some comments before accepting for publication. The following comments attached below should be addressed carefully.

The manuscript displays poor English presentations in multiple places.
Round 2
Reviewer 2 Report
Presently, the manuscript seems well-written with the proper explanations. However, in the reply to question 2, the authors claimed that the two tests are different and hence the time scale is different. Please write down the time scale for the two measurements.
